# TRAKR - A RESERVOIR-BASED TOOL FOR FAST AND ACCURATE CLASSIFICATION OF NEURAL TIME-SERIES PATTERNS

## ABSTRACT

Neuroscience has seen a dramatic increase in the types of recording modalities and complexity of neural time-series data collected from them. The brain is a highly recurrent system producing rich, complex dynamics that result in different behaviors. Correctly distinguishing such nonlinear neural time series in real-time, especially those with non-obvious links to behavior, could be useful for a wide variety of applications. These include detecting anomalous clinical events such as seizures in epilepsy, and identifying optimal control spaces for brain machine interfaces. It remains challenging to correctly distinguish nonlinear time-series patterns because of the high intrinsic dimensionality of such data, making accurate inference of state changes (for intervention or control) difficult. Simple distance metrics, which can be computed quickly do not yield accurate classifications. On the other end of the spectrum of classification methods, ensembles of classifiers or deep supervised tools offer higher accuracy but are slow, data-intensive, and computationally expensive to train and deploy. We introduce a reservoir-based tool, state tracker (TRAKR), which offers the high accuracy of ensembles or deep supervised methods while preserving the computational benefits of simple distance metrics. After one-shot training, TRAKR can accurately, and in real time, detect deviations in test patterns. By forcing the weighted dynamics of the reservoir to fit a desired pattern directly, we avoid many rounds of expensive optimization. Then, keeping the output weights frozen, we use the error signal generated by the reservoir in response to a particular test pattern as a classification boundary. We show that using this approach, TRAKR accurately detects changes in synthetic time series. We then compare our tool to several others, showing that it achieves classification performance on par with supervised deep networks on a benchmark dataset–sequential MNIST–, while outperforming all other approaches. When the samples are corrupted by noise, our approach maintains relatively high performance, while supervised deep networks show a sharp decline in performance. We also apply TRAKR to electrocorticography (ECoG) data from the macaque orbitofrontal cortex (OFC), a higher-order brain region involved in encoding the value of expected outcomes. We show that TRAKR can classify different behaviorally relevant epochs in the neural time series with high accuracy. Altogether, we show that TRAKR is a high performing tool for distinguishing patterns in complex nonlinear time-series data, such as neural recordings. With its high performance, robustness to noise, low train- and inference-time, and ease-of-use, it offers a viable alternative to more complex state-of-the art approaches, particularly for real-time applications.

## 1 INTRODUCTION

The size and complexity of neural data collected has increased greatly (Marblestone et al. (2013)). Neural data display rich dynamics in the firing patterns of neurons across time, resulting from the recurrently connected circuitry in the brain. As our insight into these dynamics increases through new recording modalities, so does the desire to understand how dynamical patterns change across time and, ultimately, give rise to different behaviors.

A lot of work in computational neuroscience over the past decade has focused on modeling the collective dynamics of a population of neurons in order to gain insight into how firing patterns are related to task variables (Márton et al. (2020); Richards et al. (2019); Yang et al. (2018); Remington et al. (2018); Kell et al. (2018); Zeng et al. (2018); Pandarinath et al. (2018); Durstewitz (2017); Chaisangmongkon et al. (2017); Rajan et al. (2016); Sussillo et al. (2015); Mante et al. (2013); Sussillo & Barak (2013); Barak et al. (2013); Sussillo & Abbott (2009)). These approaches, however, rely on fitting the whole dynamical system through many rounds of optimization, either indirectly by modeling the task inputs and outputs (Márton et al. (2020); Kell et al. (2018); Chaisangmongkon et al. (2017); Sussillo et al. (2015); Mante et al. (2013); Sussillo & Barak (2013), or directly by fitting the weights of a neural network to recorded firing patterns (Pandarinath et al. (2018); Durstewitz (2017)). Thus, these approaches can be too time- and computation-intensive for certain applications, e.g. in clinical settings where decisions need to be taken based on recordings in real-time. In these settings, neural time-series patterns need to be accurately distinguished in order to, say, detect the onset of seizures, or distinguish different mental states.

Previous approaches to classifying time series lie on a spectrum from simple distance metrics (e.g., Euclidean) to more computationally intensive approaches such as dynamic time warping (Xing et al. (2010)), ensembles of classifiers (Bagnall et al.) or deep supervised learning (Jeong (2020); Fawaz et al. (2019)). Computing simple distance metrics is fast and straightforward, but does not always yield high accuracy results because the patterns may not be perfectly aligned in time. On the other end of the spectrum, ensembles of classifiers and deep learning-based approaches (Bagnall et al.; Jeong (2020); Fawaz et al. (2019)) have been developed that can offer high accuracy results, but at high computational cost. Dynamic time warping (DTW) has been consistently found to offer good results in practice relative to computational cost (Fawaz et al. (2019); Bagnall et al. (2016); Serrà & Arcos (2014)) and is currently routinely used to measure the similarity of time-series patterns.

Previous work in reservoir computing has shown that networks of neurons can be used as reservoirs of useful dynamics, so called echo-state networks (ESNs), without the need to train recurrent weights through successive rounds of expensive optimization (Vlachas et al. (2020); Pathak et al. (2018); Vincent-Lamarre et al. (2016); Buonomano & Maass (2009); Jaeger & Haas (2004); Jaeger (a;b); Maass et al. (2002)). This suggests reservoir networks could offer a computationally cheaper alternative to deep supervised approaches in the classification of neural time-series data. However, the training of reservoir networks has been found to be more unstable compared to methods that also adjust the recurrent connections (e.g., via backpropagation through time, BPTT) in the case of reduced-order data (Vlachas et al. (2020)). Even though ESNs have been shown to yield good results when fine-tuned (Tanisaro & Heidemann (2016); Aswolinskiy et al. (2016)), convergence represents a significant problem when training ESNs end-to-end to perform classification on complex time-series datasets, and is a hurdle to their wider adoption.

Here, we propose fitting the reservoir output weights to a single time series - thus avoiding many rounds of training that increase training time and could potentially cause instabilities. We use the error generated through the output unit in response to a particular test pattern as input to a classifier. We show that using this approach, we obtain high accuracy results on a benchmark dataset - sequential MNIST - outperforming other approaches such as simple distance metrics (e.g., based on Euclidean distance or Mutual Information) and a previous approach based on echo-state networks, while performing on par with deep supervised networks. We also show that our approach is more robust to noise than other approaches, in particular deep supervised networks. At the same time, TRAKR achieves high performance while keeping training and inference time low.

We also apply our tool, TRAKR, to neural data from the macaque orbitofrontal cortex (OFC), a higher-order brain region involved in encoding expectations, and inducing changes in behavior during unexpected outcomes (Rich & Wallis (2016); Rudebeck & Rich (2018); Jones et al. (2012); Wallis (2012); Schoenbaum (2009); Burke et al. (2009); Wallis & Miller (2003); Schoenbaum et al. (1998)). This data consists of 128-channel electrocorticography (micro-ECOG) recordings obtained from the macaque OFC, including anterior and posterior areas 11 and 13, during a reward expectation task. The task was designed to understand if and how expectations encoded in OFC are updated by unexpected outcomes. TRAKR is able to distinguish three different behaviorally relevant epochs based on the neural time-series with high accuracy, revealing that OFC units differentiate between different task episodes. This shows it can be used as a reliable tool to gain further insight into the information encoded in neural circuits.

Taken together, we make the following contributions:
- We show that by fitting single time series and working with the error signal reservoirs can perform on par with supervised deep networks in time series classification
- We show that this approach, while easy to use, outperforms other commonly used approaches, such as Dynamic Time Warping (DTW), and other distance measures embraced for their simplicity
- We show that our approach is more robust to noise than other approaches
- Our approach is computationally less expensive than other approaches, achieving high performance at low train- and inference- time
- It is able to detect differences between neural patterns with high accuracy

Altogether, this shows that TRAKR is a viable alternative to state-of-the-art approaches for time series classification.

## 2 METHODS

### 2.1 MODEL DETAILS

TRAKR (Figure 1A) is a reservoir-based recurrent neural network (RNN) with $N$ recurrently connected neurons. Recurrent weights, $J$, are initialized randomly and remain aplastic over time (Buonomano & Maass (2009); Jaeger (b); Maass et al. (2002)). The readout unit, $z_{out}$, is connected to the reservoir through a set of output weights, $w_{out}$, which are plastic and are adjusted during training. The reservoir also receives an input signal, $I(t)$, through an aplastic set of weights $w_{in}$.

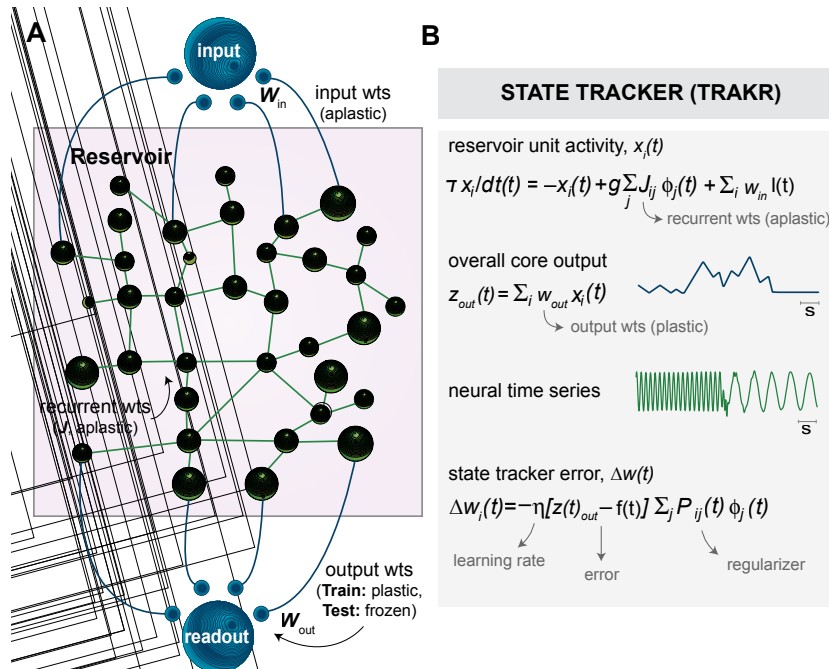

Figure 1: A) TRAKR setup overview. TRAKR consist of a reservoir connected to input and readout units via dedicated weights. Recurrent weights $J$ and input weights $w_{in}$ are aplastic. Only the output weights $w_{out}$ are subject to training. B) TRAKR equations for single unit activity, readout unit activity and error term.

The network is governed by the following equations:

$$\tau \frac{dx_i(t)}{dt} = -x_i(t) + g \sum_{j=1}^{N} J_{ij} \phi_j(t) + w_{i,in} I(t) \tag{1}$$

$$z_{out}(t) = \sum_i w_{out,i}(t)x_i(t) \tag{2}$$

Here, $x_i(t)$ is the activity of a single neuron in the reservoir, $\tau$ is the integration time constant, $g$ is the gain setting the scale for the recurrent weights, and $J$ is the recurrent weight matrix of the reservoir. The term $g\sum_{j=1}^{N} J_{ij}\phi_j(t)$ denotes the strength of input to a particular neuron from other neurons in the reservoir and $I(t)$ is the input signal (Equation 1). $z_{out}(t)$ denotes the activity of the readout unit together with the output weights, $w_{out}$ (Equation 2). In our notation, $w_{ij}$ denote the weight from neuron $j$ to $i$, and so $w_{out,i}$ means the weight from $i^{th}$ unit in the reservoir to the readout unit. $\phi$ is the activation function given by:

$$\phi_i(t) = tanh(x_i(t)) \tag{3}$$

We use recursive least squares (RLS) to adjust the output weights, $w_{out}$ during training (Haykin, Simon S. (1996)). The algorithm and the update rules are given by:

$$\Delta w_{out,i}(t) = -\eta(z_{out}(t) - f(t))\sum_j P_{ij}(t)\phi_j(t) \tag{4}$$

$$w_{out,i}(t) = w_{out,i}(t-1) + \Delta w_{out,i}(t) \tag{5}$$

Here, $\eta$ is the learning rate, $f(t)$ is the target function, and the term $\sum_j P_{ij}(t)\phi_j(t)$ acts as a regularizer where $P$ is the inverse cross-correlation matrix of the network firing rates. For details on setting hyperparameters, see Appendix A.

## 2.2 ADJUSTING RESERVOIR DYNAMICS

During training, the output weights, $w_{out}$, are optimized using RLS based on the instantaneous difference between the output, $z_{out}(t)$, and the target function, $f(t)$. This optimization is performed in one shot (without the need for multiple optimization rounds). Here, we use the reservoir to autoencode the input signal, thus $f(t) = I(t)$. The instantaneous difference gives rise to an error term, $E(t)$, calculated as:

$$E(t) = \sum_{i=1}^{N} \Delta w_{out,i}(t) \tag{6}$$

## 2.3 OBTAINING THE ERROR SIGNAL

After training, the output weights, $w_{out}$ are frozen. The test pattern is fed to the network via the input, $I(t)$, and the network is iterated to obtain the error, $E(t)$ over the duration of the test signal. The error, $E(t)$ is computed as the difference between the test signal and the network output (Equation 6). The error may vary depending on the similarity of a given test signal to the learned time series. The error is used as input to a classifier.

## 2.4 CLASSIFICATION OF THE ERROR SIGNAL

The error, $E(t)$, is used as input to a support vector machine (SVM) classifier with a Gaussian radial basis function (rbf) kernel. The classifier is trained using 10-fold stratified cross-validation. The same classifier and training procedure was used in comparing the different approaches. Naive Bayes as well as the neural network-based approaches (multilayer perceptron (MLP) and time warping invariant echo state network (ESN)) are directly used as classifiers, again with 10-fold stratified cross-validation. Accuracy and area under the curve (AUC) are computed as a measure of classification performance. MacBook Pro CPU was used for all the comparisons.

### 2.5 NEURAL RECORDINGS

#### 2.5.1 TASK DESIGN

Neural recordings were obtained from the macaque OFC using a custom designed 128-channel micro-ECOG array (NeuroNexus), with coverage including anterior and posterior subregions (areas 11/13). During preliminary training, the monkey learned to associate unique stimuli (natural images) with rewards of different values. Rewards were small volumes of sucrose or quinine solutions, and values were manipulated by varying their respective concentrations.

The behavioral task design is shown in (Figure 4A). During the task, the monkey initiated a trial by contacting a touch-sensitive bar and holding gaze on a central location. On each trial, either one or two images were presented, and the monkey selected one by shifting gaze to it and releasing the bar. At this point, a small amount of fluid was delivered, and then a neutral cue appeared (identical across all trials) indicating the start of a 5s response period where the macaque could touch the bar to briefly activate the fluid pump. By generating repeated responses, it could collect as much of the available reward as desired. There were two types of these trials. Match (mismatch) trials were those where the initial image accurately (did not accurately) signal the type of reward delivered on that trial. Behavioral performance and neural time series were recorded in 11 task sessions across 35 days. For further details on trials and data preprocessing, see Appendix B.

## 3 RESULTS

### 3.1 DETECTING PATTERN CHANGES IN SYNTHETIC TIME SERIES

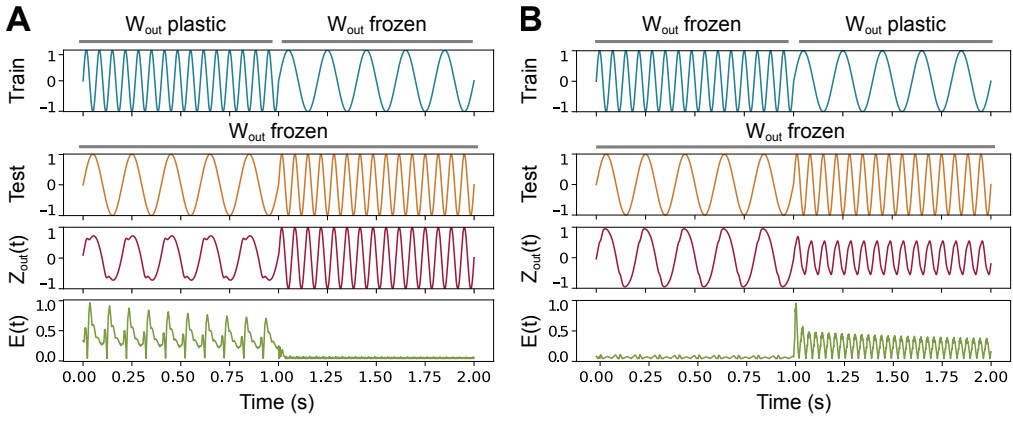

Figure 2: A) (Blue) $w_{out}$ plastic for a 15 Hz $sin$-function, and frozen for a 5 Hz rhythm. (Orange) Test pattern with the same frequencies but the signal order reversed. (Red) TRAKR output. (Green) The error signal, $E(t)$, is showing an increase for the part of the test pattern which was not learned during training. B) Similar to A but $w_{out}$ were plastic during the second half of the training signal (5Hz rhythm).

First, we trained TRAKR on idealized, synthetic signals using $sin$-functions of two different frequencies (Figure 2). Reservoir output weights were fitted to the signal using recursive least squares (RLS; see subsection 2.1). In Figure 2A, the network was trained on the first half of the signal (blue) while the output weights, $w_{out}$, remained frozen during the second half. Then with $w_{out}$ frozen, a test signal (orange) was fed to the reservoir. The network output, $z_{out}(t)$, in red and the error signal, $E(t)$, in green are depicted in Figure 2. The network correctly detects the deviation of the test pattern (orange, 1st half of the signal) from the learned pattern (blue, 1st half of the signal), which results in an increase in the error signal (green, 1st half of the signal, Figure 2A). The second half of the test signal (orange) aligns with the trained signal (blue, 1st half) and thus yields no error (green, 2nd half). In Figure 2B, the order of the training procedure was reversed in that output weights remained frozen for the first half of the signal (blue) and were plastic during the second half. As expected,

the increase in the error signal (green) now occurs during the second half of the test signal (orange). Thus, TRAKR correctly detects, via the error signal $E(t)$, when a new frequency pattern occurs in the test signal that deviates from the trained pattern.

## 3.2 CLASSIFYING DIGITS - SEQUENTIAL MNIST

We then applied TRAKR to the problem of classifying the ten digits from sequential MNIST, a benchmark dataset for time-series problems (Le et al. (2015); Kerg et al. (2019)).

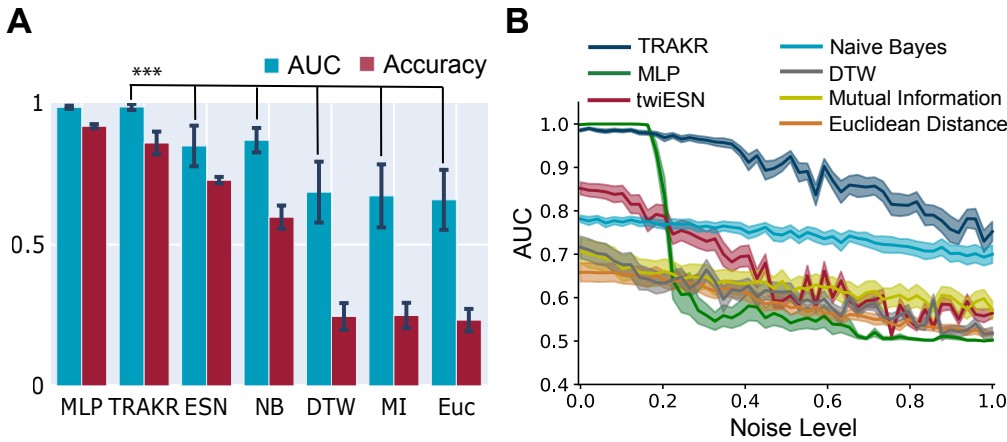

Figure 3: Classification performance on the sequential-MNIST dataset Le et al. (2015). **A** MLP: Multi-layer perceptron (as in Fawaz et al. (2019)); ESN: echo-state network (twiESN, as in Fawaz et al. (2019)); NB: Naive Bayes; DTW: Dynamic Time Warping (as in Rakthanmanon et al. (2012)); MI: Mutual Information; Euc: Euclidean distance metric. TRAKR performs on par with MLPs, while outperforming all other methods ($99\%\ AUC$; $***: p < 0.001$, Bonferroni-corrected) **B** Classification performance under increasing amount of noise. TRAKR performance declines gradually with noise intensity, while MLP performance degrades abruptly at higher noise levels. Chance level is at $10\%$.

For training, we used the entire dataset of 1000 sequential MNIST digits including 100 samples for each digit (0-9). We fed each sequential digit (28 x 28 pixel image flattened into a vector of length 784) as a one-shot training signal to TRAKR. Reservoir output weights were again fitted to a single digit using recursive least squares (RLS; see subsection 2.1). After fitting TRAKR to one time series corresponding to one of the samples of a particular digit, we froze the output weights and fed all the other digits as test samples to TRAKR. We obtained an error signal, $E(t)$, from every test sample, with the magnitude of the error varying depending on the similarity with the learned digit. The error signal was then fed into a classifier which was trained to differentiate the digits based on the error terms (see subsection 2.4 for more details). We repeated this procedure for all digits and samples in the dataset to obtain the averaged classification performance for TRAKR (Figure 3A).

We found that TRAKR achieves a performance of $AUC = 99\%$ on this dataset (Figure 3A). We compared our approach with other commonly used methods for the classification of time series. We compared our results against supervised deep neural networks (MLP; as in Fawaz et al. (2019)), a recent echo-state network based approach (twiESN; as in Fawaz et al. (2019)), distance measures such as Dynamic Time Warping (DTW; Rakthanmanon et al. (2012)), Euclidean distance Euc), Mutual information (MI), and a naive Bayes classifier (NB). With the exception of MLPs, which showed performance on par with TRAKR, we found that all other approaches performed significantly worse than TRAKR ($p < 0.001$; see subsection 2.4 for further details).

We also tested the performance of TRAKR under different noise levels added to training digits (Figure 3B, Appendix C). We again compared TRAKR against all the other approaches and found TRAKR to perform the best, particularly at higher noise levels: performance decays gradually as the noise is increased and is at $AUC = 75\%$ even at higher noise levels ($\sigma = 1$). In particular, we found TRAKR performs better than MLPs at higher noise levels.

Table 1: Computational cost compared

| Method | Training time (ms) $[mean \pm sd]$ | Inference time (ms) $[mean \pm sd]$ |
|--------|-----------------------------------|-------------------------------------|
| MLP | 110200.00 $\pm$8700.00 | 92.50 $\pm$18.40 |
| twiESN | 105500.00 $\pm$5200.00 | 250.35 $\pm$10.25 |
| **TRAKR** | (42.10 $\pm$1.21)+svm train time | (45.40 $\pm$1.38)+svm test time |
| DTW | (0.30 $\pm$0.20)+svm train time | (0.30 $\pm$.20)+svm test time |
| MI | (0.20 $\pm$0.10)+svm train time | (0.20 $\pm$0.10)+svm test time |
| Euc | (0.02 $\pm$0.01)+svm train time | (0.02 $\pm$0.01)+svm test time |
| NB | 252.60 $\pm$10.15 | (5.22 $\pm$1.41) |
| SVM | 15200.00 $\pm$3500.00 | 80.60 $\pm$20.30 |

We also measured training and inference time, comparing TRAKR to the other approaches introduced above (Table 1). We found that training time for TRAKR is situated at the lower end of the spectrum, slower than it takes to obtain a measure of the distance between two traces using DTW, MI or Euc, but significantly faster than it takes to train the MLP or twiESN. While it does require upfront fitting, our approach has the advantage that it does not require multiple rounds of optimization (like MLP or twiESN) because the signal is fit in one shot (see subsection 2.2 for details). This yields relatively fast training time.

After fitting, TRAKR can detect deviations from the learned signal in real-time. The inference time of our approach again lies in between computationally relatively more expensive approaches such as MLP and twiESN and less intensive approaches such as DTW, MI and Euc. We also measured the train and inference time of the classifier we use (SVM) and found that it does not significantly impact training or inference time; other, relatively more simple classifiers such as kNN (k-Nearest Neighbors) may also be used instead, for further gains in speed. We compared all of these approaches under the same conditions (see subsection 2.4). Also, see Appendix D for details on calculation of training and inference time.

### 3.3 PERFORMANCE ON NEURAL TIME SERIES RECORDED FROM THE MACAQUE OFC

The OFC is involved in encoding and updating affective expectations. We used a behavioral task designed to study the neural mechanisms of how such expectations are encoded in the OFC of primates and how they may guide behavior under different conditions. The goal here was to determine whether TRAKR can be used to reliably distinguish complex neural time series patterns (Figure 4A; see also subsubsection 2.5.1 for more details).

A recording from a single electrode is shown together with three different behaviorally relevant epochs (rest, choice and reward periods; Figure 4B). We compared the performance of TRAKR with all other previously introduced methods on the task of trying to distinguish the neural time series patterns in these three epochs. It is important to note that we did not know a priori whether there is enough information encoded in the OFC recordings to allow for differentiating the three epochs. Thus, a high performing classifier can offer clues here as to what type of information is encoded in the brain.

We trained TRAKR on the neural time series corresponding to rest period from a particular trial, and used the other complete trials as test signals to obtain the error, $E(t)$. The error signal was used as input to a classifier. We repeated this procedure for all trials in the dataset to obtain the averaged classification performance. We also calculated the Fast Fourier transform (FFT) of the signals and obtained the magnitude (power) in the $\alpha$ $(0 - 12Hz)$, $\beta$ $(13 - 35Hz)$, and $\gamma$ $(36 - 80Hz)$ bands within the 3 epochs. We compared TRAKR against this FFT based classification too.

Again, with the exception of MLP which performed on par with TRAKR, we found that TRAKR outperformed all the other methods ($AUC = 94\%$; $p < 0.001$; Figure 4C). TRAKR was able to distinguish the three epochs with high accuracy based on the neural signal, showing that there is enough information in the OFC to differentiate these patterns. It is important to note that based on the performance of lower performing approaches, such as DTW or Euc, one might have (wrongly) concluded that the OFC lacks enough information to solve this task. Overall, with its high per-

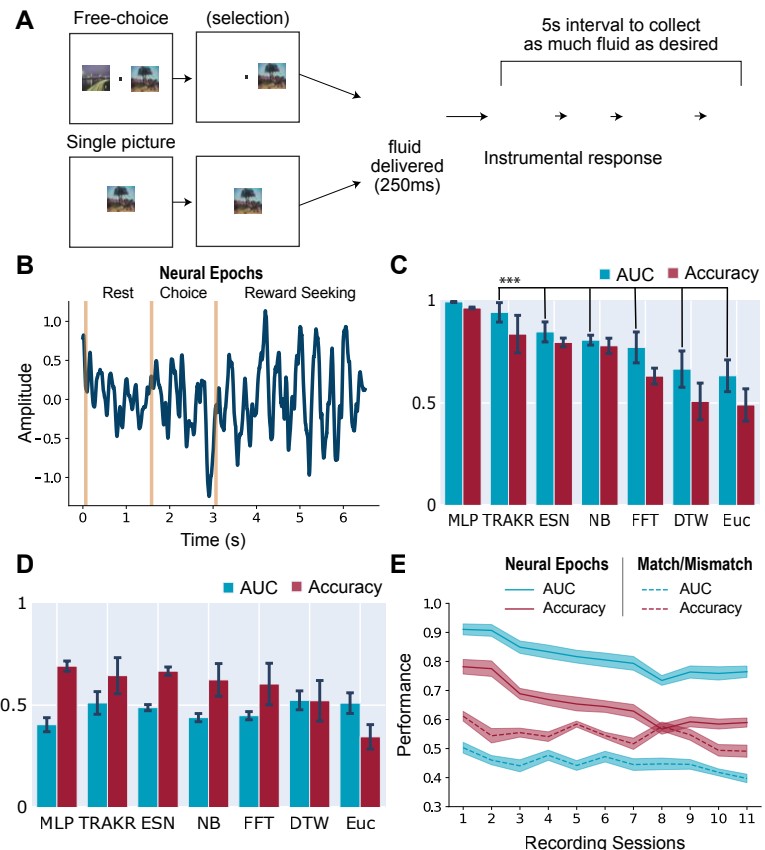

Figure 4: A) Neural task design (see subsubsection 2.5.1 for detailed description). B) Example neural time series from a single trial, with three behaviorally relevant epochs (rest, choice and instrumental reward seeking period). C) MLP: Multi-layer perceptron (as in Fawaz et al. (2019)); ESN: echo-state network (twiESN, as in Fawaz et al. (2019)); NB: Naive Bayes; DTW: Dynamic Time Warping (as in Rakthanmanon et al. (2012)); MI: Mutual Information; Euc: Euclidean distance metric. TRAKR performs on par with MLP, while outperforming all other methods in classifying the different neural epochs ($***: p < 0.001$, Bonferroni-corrected; chance-level at 33%). D) All methods perform at chance-level (50% $AUC$) in distinguishing match/mismatch trials, suggesting that the OFC does not encode enough information to distinguish between the two conditions. E) Classification performance (TRAKR) decreases over 11 recording sessions (35 days).

formance, TRAKR can be used as a reliable tool to differentiate time series patterns and generate hypotheses on the type of information represented in neural circuits.

We also investigated whether any of the methods could distinguish between match and mismatch trials based on the OFC signals (Figure 4D). For this purpose, we trained TRAKR on the neural time series corresponding to choice period from a particular trial, and used the other complete trials as test signals to obtain the error, $E(t)$. We found that all methods performed at chance-level, indicating there is not enough information in the OFC recordings to differentiate the two. We also observed that the classification performance for the three epochs degrades over days (Figure 4E; blue & red solid lines), while that for match/mismatch trials consistently stays around chance-level (Figure 4E; blue & red dotted lines).

Lastly, we visualized different electrodes in the space spanned by the first three principal components of the reservoir's activations (Appendix E). We fitted the reservoir to signal obtained from a particular electrode, froze the output weights, and projected other electrodes onto the first three principal components of reservoir activity. We found that electrodes trace out different paths in reservoir space. Thus, as a proof-of-concept, projections onto reservoir space can be used to visually inspect

the similarity of recordings from different electrodes, and identify differences that may be indicative of functionally meaningful sub-groupings that represent functionally coherent modules in the brain.

## 4 DISCUSSION

We have shown that TRAKR can distinguish time series patterns with high accuracy. TRAKR outperforms other approaches in classifying time-series data on a benchmark dataset, sequential MNIST, and in differentiating neural time series signals obtained from recordings in the macaque OFC. It performs on par with supervised neural networks (MLPs), while outperforming other types of echo state networks (twiESN) and commonly used distance measures such as Dynamic Time Warping (DTW). Meanwhile, we found that our approach is more robust to noise than other approaches, in particular supervised neural networks, and offers good performance in terms of training and inference time.

We found that TRAKR offers high accuracy at relatively lower training and inference times than other approaches with comparable accuracy such as supervised multi-layer neural networks. While TRAKR relies on a classifier on top of the error traces to perform the classification task, other classifiers such as kNN may be used to maximize training and inference speed, instead of the SVM classifier employed here. All approaches were compared in the same environment, but all neural network-based approaches may equally benefit from further gains in training and inference speed by optimising the code for deployment on GPUs. For this purpose, TRAKR can readily be implemented in JAX, a high-performance computing framework (Bradbury et al. (2018).

Meanwhile, other approaches based on echo-state networks, such as twiESN (Fawaz et al. (2019)), performed worse than TRAKR and showed higher train and inference times. This suggests our contribution is key to making reservoir networks a viable alternative to state-of-the-art approaches in time series classification.

While TRAKR and most other methods could distinguish three behaviorally relevant epochs based on the OFC signal, none of the methods were able to accurately distinguish match and mismatch trials. This indicates that there is enough information in the OFC to distinguish the three task periods, but not enough to differentiate match and mismatch trials. It is possible that receiving a better or worse reward than expected affected the neural signal in distinct/opposite ways, such that the effect was cancelled out on average. It is also possible that the difference in neural time-series patterns was only discernible if the reward was maximally different (much better or worse than expected). In the current task design, there were 4 different levels of reward (flavors) that the macaque associated with different pictures (subsubsection 2.5.1). The number of trials in which obtained was maximally different from expected reward was low and possibly not sufficient for accurate classification. Another possibility, supported by our results and corroborated by several studies (Stalnaker et al. (2018); McDannald et al. (2014); Takahashi et al. (2013); Kennerley et al. (2011)), is that OFC neural activity signals reward *values* but not reward *prediction errors*, which instead are mediated through the ventral tegmental area (VTA) in the midbrain.

We found that the classification performance decreased over recording sessions. This could mean that the difference between task epochs being classified decreased because of increased familiarity with the task. That is less likely, however, because the subject was well-trained prior to recordings. Instead, since the signal was recorded over a period of 35 days, the decrease in the classification performance could be a result of degrading signal quality, perhaps due to electrode impedance issues (Kozai et al. (2015a;b); Holson et al. (1998); Robinson & Camp (1991)).

## 5 CONCLUSION

There is a need for and strong interest in tools for the analysis of time-series data (Bhatnagar et al. (2021)). We show that TRAKR is a fast, accurate and robust tool for the classification of time-series patterns. Through its ease of use and low training and inference time, it is particularly suited for real-time applications where accurate decisions need to be made quickly and signal degradation or other artifacts necessitate frequent re-calibration, such as in clinical settings. TRAKR can also be used to distinguish neural time series patterns in the brain, shedding light on the information encoded in neural circuits and thus generating hypotheses for new experiments.

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

## A  TRAKR HYPERPARAMETERS

The recurrent weights $J_{ij}$ are weights from unit $j$ to $i$. The recurrent weights are initially chosen independently and randomly from a Gaussian distribution with mean of 0 and variance given by $g^2/N$. The input weights $w_{in}$ are also chosen independently and randomly from the standard normal distribution.

An integration time constant $\tau = 1ms$ is used. We use gain $g = 1.2$ for all the networks.

The matrix $P$ is not explicitly calculated but updated as follows:

$$P(t) = P(t-1) - \frac{P(t-1)\phi(t)\phi'(t)P(t-1)}{1 + \phi'(t)P(t-1)\phi(t)}$$

The learning rate $\eta$ is given by $\dfrac{1}{1 + \phi'(t)P(t)\phi(t)}$.

The number of units used in the reservoir is generally $N = 30$.

## B  MACAQUE TRIAL DETAILS & DATA PRE-PROCESSING

Each trial was approximately 6.5s long, including different behaviorally relevant epochs and cues. The macaque performed approximately 550 trials within each task session ($mean \pm sd : 562 \pm 72$). Of note, 80% of the trials were match trials within each task session.

ECoG data were acquired by a neural processing system (Ripple) at $30kHz$ and then resampled at $1kHz$. The 128-channel data were first z-score normalized. Second-order butterworth bandstop IIR filters were used to remove 60Hz line noise and harmonics from the signal. We also used second-order Savitzky-Golay filters of window length 99 to smooth the data and remove very high frequency juice pump artifacts ($> 150Hz$). For most of the analysis here, we used the average of the 128-channel time series as an input to TRAKR.

## C  DETAILS ON SEQUENTIAL MNIST CLASSIFICATION

For measuring noise robustness, we added random independent Gaussian noise to the training digits ($\mu = 0$ and varying standard deviation ($\sigma$)). The actual noise that was added (noise levels as depicted in Figure 3B) can be calculated as $\sigma * 255$, with $\sigma \in [0,1]$. The number 255 represents the maximal pixel value in the sequential digits.

## D  CALCULATION OF TRAINING AND INFERENCE TIME

The calculation of training time for TRAKR includes the time taken to train one sequence through TRAKR along with 10-fold cross validation time using SVM. For all the other methods, it is also the time taken to train using 10-fold cross validation. For MLP, the training is done using Keras which is optimized, whereas TRAKR has further room for optimization by implementing it in JAX etc.

The calculation of inference time for TRAKR includes the time taken to feed one test sequence into TRAKR and making predictions on a test set using SVM. For all the other methods, the time includes making predictions on the same test set.

## E  VISUALIZATION OF RESERVOIR ACTIVATIONS

Single electrode recordings were projected into the space spanned by the first three principal components of reservoir activations. The four electrodes trace out different trajectories in reservoir space, suggesting they capture potentially different neural dynamics, as shown in the figure below.

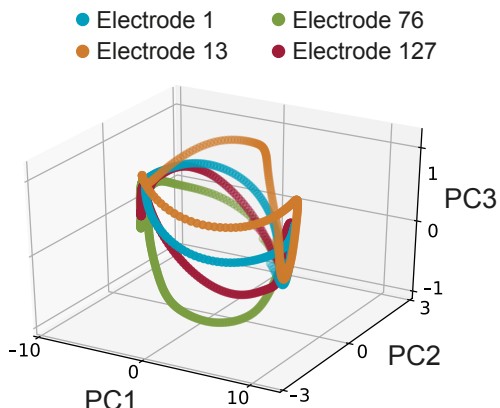

