# OpenReview forum: "TRAKR – A reservoir-based tool for fast and accurate classification of neural time-series patterns"
_ICLR.cc/2022/Conference — ICLR 2022 Submitted_

### Official Review · Reviewer_dzZZ · 2021-10-29

**Correctness:** 3
**Technical Novelty And Significance:** 3
**Empirical Novelty And Significance:** 2
**Recommendation:** 6
**Confidence:** 3

**Main Review:**

Strengths:

1. The manuscript is straightforward and well-written
2. The notion of using the error signal as the input to a classifier as new and potentially interesting
3. The classification results are possibly compelling

Major issues:

1. A central claim of the paper is that echo state networks are both performant and computationally efficient. Firstly, to properly contextualize the accuracy of their technique, the authors should compare TRAKR against more state of the art methods. For instance, the author should consider implementing some of the methods from the Fawaz review they cite. Moreover, while deep-learning methods may be more expensive to train, they can be used as a more up-to-date benchmark. Additionally, this could present the authors with an opportunity to compare the performance of TRAKR with deep learning methods as a function of the size of the training dataset (I suspect TRAKR might look more favorable here).

2. Another major claim is that TRAKR is more "lightweight" than ensemble or deep-learning methods. Deep learning methods, especially for large network sizes, are particularly data-hungry in the training phase. However, inference with deep networks can be extremely fast, to the point of being sufficient for real-time applications. If one were to compare the inference time of a deep network on a GPU vs TRAKR, it is possible that TRAKR is not significantly faster than a better-performing deep network. This must be clarified since computational efficiency is touted as a major feature of TRAKR.

Minor issues:

1. Figure 4 what do the error bars repesent

**Summary Of The Paper:**

Here, the authors propose a reservoir-computing based framework for time-series applications. The proposed methodology involves training the output units of the echo state network to recapitulate, or autoencode, a test signal. Next, time-series are fed to the network and the error signal between the time-series and the network's response is used as input to an SVM that is used for classification.  Effectively the error signal is used as a distance metric.

The authors report that the network performs well on sequential MNIST and decoding of neural data. Additionally, they argue that a key advantage of their approach is that it is computationally lightweight – both training and inference are fast.

**Summary Of The Review:**

While the manuscript is well-written and the ideas presented are fairly straightforward, the authors need to address the primary competitor to TRAKR, deep nets, head-on. First, it is likely that deep nets are on par if not better-performing than TRAKR, which the authors readily admit.  However, the authors also need to perform analysis of inference speed with deep networks.  If deep nets can run almost as quickly as TRAKR, then this complicates a central pillar of the paper.

---

> ### Author Response · Authors · 2021-11-22
> **Response to reviewer's comments**
>
> We appreciate the time the reviewer took to assess our work, and will reply to each point individually below. We agree with the reviewer on the main strengths of our work.
>
> 1) We appreciate this suggestion, and have now benchmarked TRAKR against other neural-network approaches also employed in Fawaz et al (2019). We used the implementation of the MLP and time-warping invariant ESN exactly as in Fawaz et al. (2019).
> Comparing our approach to time warping invariant ESN (Tanisaro & Heidemann, 2016), we find it yields lower performance (73% Accuracy) compared to our approach, at relatively higher inference time (~250ms vs. 40ms for TRAKR without SVM). Comparing to MLP (multi-layered perceptron), using the same configuration as in Fawaz et al. (2019), we find it achieves similar performance at higher inference time (100ms vs. 40ms for TRAKR without SVM). Also, we found training time for MLP to be much higher compared to Trakr (110s for MLP vs. 15s for TRAKR including SVM). We have now revised Figs 3 & 4 as well as the accompanying text in the Results section to reflect these results.
> Overall, we believe these results will have a true impact for practical use of TRAKR for time-series classification tasks in a real-world setting.
>
> 2) We thank the reviewer for this clarifying question. We have found the training time for TRAKR to be 40ms + svm train time (15 s) and the inference time to be 40ms +svm test time (80ms) . This compares to 110s train time and 100ms inference time for the MLP, and 105s training time and 250ms inference time for the ESN approach (twiESN), as further explained above (in response to point 3). Although we have used SVM for the classification of error signals here, other relatively more simple classifiers such as kNN (k-Nearest Neighbors) may also be used instead, for further gains in speed.
> We compared all approaches on a Macbook Pro CPU. As you mention, these approaches will surely gain from optimizing them for use on GPUs, but they will all benefit equally. We are in fact currently working on a JAX-based implementation of TRAKR, which will further decrease training and inference time, and we are planning to include it with the final submission.
> While we agree that inference time is crucial, we also believe that training time is of concern in many real-world applications, especially in the medical realm: recording electrodes degrade/move with time and biological signals shift, requiring constant re-training of classifiers. TRAKR performs well on both fronts. We believe this will have a big impact for the practical use of our methods in real-world settings.
> We clarify these points in the revised manuscript, as well as include a revised Table 1 comparing all the methods and including another table to compare training time.
>
> Minor issues: 1) The reported AUC represents the grand average across all possible reference signals, classes and classification folds. The error bars in Fig.3 & 4 represent the standard deviation across all those parameters.

---

> > ### Comment · Reviewer_dzZZ · 2021-11-29
> > **Increased score from 5 to 6 to reflect the author's revisions**
> >
> > The authors have clearly addressed all of my concerns, so I am increasing my score from 5 to 6.
> >
> > Agreed with the other reviewer that the new benchmarks are a nice addition to the paper, and I think the comparisons the authors make are a fair and make a good case for the utility of TRAKR.
> >
> > Looking forward to the JAX implementation!

---

> ### Author Response · Authors · 2021-11-22
> **Manuscript Revision Uploaded**
>
> We thank the reviews for their time and insightful comments. We have uploaded a revised manuscript based on their comments. Concretely, the following updates have been made:
>
> - The abstract has been revised based on reviewer suggestions, and an updated manuscript provided.
> - We compared the performance of TRAKR with 2 additional supervised learning methods: a deep learning based approach (multi layer perceptron (MLP) as in Fawaz et al 2019) and an echo-state network approach (time warping invariant echo state network (twiESN) as in Fawaz et al 2019). We show that TRAKR performs on par with MLP and outperforms the previous ESN approach in both the sequential MNIST dataset (Figures 3A and 3B) as well as the macaque OFC neural recordings (Figures 4C and 4D). We also revised the text related to these in sections 3.2, 3.3 and 4.
> - Updated Figures 3A, 3B, 4C and 4D as mentioned and updated text in the Results section to reflect those changes. Moved former Figure 5 to the appendix and revised the accompanying text at the end of Results.
> - Updated all Figure captions
> - We revised Table 1, which now shows both training and inference times for all methods, including the train and inference time for the SVM classifier used for TRAKR and other approaches.
> - We re-ran the analyses with a different DTW implementation (based on Rakthanmanon et al 2012) which is described as faster and more accurate than the previously used implementation (FastDTW). We adjusted the captions of Figures 3 and 4 to reflect this change.
> - In response to reviewers’ comments, we now used 10-fold stratified cross validation throughout, instead of leave-one-out cross validation as before, as further explained in section 2.4.
> - Moved some details to the appendix sections; specifically:
> --Details on the length of macaque trials
> --Data preprocessing of macaque neural signals
> --Details on the noise added to seq- MNIST digits
> --Former figure 5 and its caption as text

---

### Official Review · Reviewer_xpen · 2021-10-29

**Correctness:** 3
**Technical Novelty And Significance:** 2
**Empirical Novelty And Significance:** 2
**Recommendation:** 6
**Confidence:** 3

**Main Review:**

Strengths: The method is reasonably presented. The method seems novel, and valuable because it requires very little training to create a representation of the input that can be used for further tasks.

Weaknesses:
The experiments proposed in the paper do not seem sufficient to assess the strengths of the proposed method. The proposed method is not compared against other reservoir-based methods, against random-projection methods, or against deep neural networks.
___
Sinusoid toy dataset:
1. The frozen section of the train set is confusing. Why include this section in the train set if the learned weights are frozen ? Wouldn't it be clearer to remove the frozen section of the training set ?

MNIST experiment:

2. MNIST is composed on images, which are not naturally represented as sequential time-series. Why use MNIST as a dataset for time-series classification ?
3. Why is the comparison limited to the four baseline methods? Why not include other models known to be good at classifying MNIST (such as CNNs)? Why not include other methods based on reservoir computing ? Why not include other random-projection methods ?
4. How were these baseline methods applied? Were the distance metrics used as input of the RBF kernel? How is Naive Bayes used in conjunction with the SVM ?
5. Is the classification task a one-vs-one, one-vs-all, or multinomial task? If multinomial, how is the AUC computed ?
6. How are the error bars computed in Figure 3A ? How is the shaded area computed in Figure 3B ? Assuming this is some sort of standard deviation around a mean value, why does the mean AUC for TRAKR fluctuates more than the standard deviation (e.g. trough at noise level = 0.7)?
7. How much time does the method requires to (a) train the reservoir, (b) train the SVM, (c) predict a sample ? Which computational time is reported in Table 1?

The OFC experiment raises the same questions as the MNIST experiment in terms of choice of the baseline methods. Additionally, it is not clear how the task of predicting behavioral states is useful to understand how the brain works. The TRAKR method does not seem to give particularly interpretable representations. The TRAKR+PCA embedding presented in Figure 5 also lacks a comparison with other embeddings, such as PCA directly on the electrode signals.

**Summary Of The Paper:**

The paper proposes a new method to classify neural time-series. The method is based on a recurrent reservoir, which includes a black box function (here a fixed random RNN) and a linear readout. The black box function produces an state vector at each time point of an input time-series. This state vector is then transformed with a linear readout, trained to reconstruct the input signal. In other words, the reservoir is an auto-encoder with a fixed random RNN encoder and a linear decoder trained to minimize the reconstruction error. The method then uses the reconstruction error of the reservoir as a representation of the signal, to be used in a RBF-kernel SVM classifier and perform a given task.

On a simple toy example, the authors show that the reservoir can learn to reconstruct a sinusoid signal, and that it fails to reconstruct a sinusoid with a frequency different than during training. This failure can be considered useful to identify regimes different that the ones seen during training. The method is then used on a small MNIST datasest, training the reservoir on one sample, and then training an SVM classifier on all samples based on the reconstruction error of the reservoir. The method is then compared with four baseline methods. Finally, the method is applied to neural recordings of a macaque OFC, training the reservoir on one epoch, and then training a classifier to predict behavioral states.

**Summary Of The Review:**

The comparison with other methods seems rather incomplete.

---

> ### Author Response · Authors · 2021-11-22
> **Response to Reviewer's comments**
>
> We appreciate the reviewer’s detailed summary and suggestions, and will address the points raised individually below.
>
> 1) We appreciate the reviewer’s remark. It is true that the signal will not be used to adjust the readout weights during the time that they are frozen. But for the sake of understanding how the various signals we compare relate to each other in terms of network time, we thought it was clearer not to remove the latter half of the training signal. If you do still think it would aid in clarity to revise this figure, we can include an improved version in the final submission.
>
> 2) Sequential MNIST is a challenging dataset for sequential information (Le, Jaitly & Hinton, 2015). Networks have to predict the category of the image only after seeing all 784 pixels, creating a long-term dependency problem. It is still used to compare performance on sequential data in more recent publications (Kerg et al, NeurIPS 2019) and hence presents itself as a good common benchmark for methods designed for sequential tasks.
>
> 3) We found these four methods represent a good range of different methods commonly used on this data, varying in computational intensity. We found it important to compare our method to other approaches with similar ease-of-use and inference speed.
> As pointed out in the Abstract/Introduction, we propose TRAKR to sit at the sweet spot of relatively high accuracy at low computational intensity. As explained above (in response to reviewer 1, point 6) and copied here for convenience, DTW is arguably the most important method to compare against:
> Dynamic time warping (DTW) is a state-of the art technique for the classification of time series signals (“DTW when used with a NN classifier has been shown to be a very strong baseline” (Fawaz et al. 2019; Bagnall et al 2017); “At a minimum, it is clear that 1NN-DTW will get you within 98% of the best accuracy possible, in the first five minutes” (Eamonn Keogh, https://www.cs.unm.edu/~mueen/DTW.pdf)). We found this technique, in its optimised implementation, to yield lower results than TRAKR on both datasets.
> Upon your suggestion, we have also compared TRAKR against other more compute-heavy approaches, such as ESNs and DNNs. As explained above (reviewer 1, point 5) and copied here for convenience, we have found the following:
> We have now compared our approach to other ESN and DNN models. Comparing our approach to time warping invariant ESN (Tanisaro & Heidemann, 2016), as also further explained in 2, we find it yields lower performance (73% Accuracy) compared to our approach, at relatively higher inference time (~250ms vs. 40ms for TRAKR alone without the SVM). Comparing to MLP (multi-layer perceptron), using the same configuration as in Fawaz et al. (2019), we find it achieves similar performance at higher inference time (100ms vs. 40ms for TRAKR alone without SVM). Also, we found training time for MLP to be much higher compared to TRAKR (110s for MLP vs. 15s for TRAKR with SVM). We have made changes to Figs 3&5 and the accompanying text in the Results section to reflect these new insights.
>
> 4) As the reviewer correctly points out, the distance metrics (Euclidean, MI) and measures (DTW) are used as input to a SVM classifier. Naive Bayes (NB) as well as the neural network-based approaches above (ESN, MLP) are directly used as classifiers. We agree that we should have made this more clear in the manuscript, and we have revised the Methods section as well as Table 1 to clarify this point.
>
> 5) The classification is one-vs-all. As mentioned in the reply to reviewer 1 (point 2), we currently compute performance across all possible reference signals and average the performance. As such, the reported AUC represents the grand average across all possible reference signals, classes and classification folds. We have now revised the figure legends accordingly.
>
> 6) As mentioned in response to the reviewer’s question above (5), the reported AUC represents the grand average across all possible reference signals, classes and classification folds. The error bars in Fig. 3 & 4 represent the standard deviation across all those parameters. We have now made changes to the figure legends to address this point.
> The shaded error bar in Fig 3B similarly represents standard deviation across all of the parameters above, for different noise levels. However, it does not include different initializations of the network. The fluctuation of the AUC around the 0.7 noise level may be due to a computation of the random noise and network configuration. Running it for different network initializations should remove this fluctuation. We have now updated this figure to address this point.

---

> > ### Author Response · Authors · 2021-11-22
> > **Response to Reviewer's comments**
> >
> > 7) Thank you for your clarifying question. We have found the training time for TRAKR to be 40ms + svm train time (15 s) and the inference time to be 40ms +svm test time (80ms) . This compares to 110s train time and 100ms inference time for the MLP, and  105s training time and 250ms inference time for the ESN approach (twiESN), as further explained above (in response to point 3). Although we have used SVM for the classification of error signals here, other relatively more simple classifiers such as kNN (k-Nearest Neighbors) may also be used instead, for further gains in speed.
> > We have now included a revised Table1 including timings for the additional methods compared. We have now separated training and inference time, and also provided separate estimates for the classifier we used.
> >
> >
> > The calculation of training time for TRAKR includes the time taken to train one sequence through TRAKR along with 10-fold cross validation time using SVM. For all the other methods, it is also the time taken to train using 10-fold cross validation. For MLP, the training is done using Keras which is optimized, whereas TRAKR has further room for optimization by implementing it in JAX etc.
> > The calculation of inference time for TRAKR includes the time taken to feed one test sequence into TRAKR and making predictions on a test set using SVM. For all the other methods, the time includes making predictions on the same test set.

---

> > > ### Author Response · Authors · 2021-11-22
> > > **Response to Reviewer's comments**
> > >
> > > We would like to point out that in reporting results we do not currently choose a particular baseline, but instead average across all possible reference signals (as further explained in response to reviewer 1, point 2, and reposted here for convenience):
> > > Currently, we do not handpick a particular reference, but instead average our results (Figs 3 & 4) across all possible references. To be fair in comparison, we also do this with other distance metrics and measures such as DTW (which is non-symmetric).
> > > In the future, the selection of a particular reference signal could be treated as a hyperparameter, to be optimized on a smaller validation set. It is important to note, though, that one can already achieve good results by picking a reference ad hoc: the variance across different references is relatively small (TRAKR error bar, Figs 3 & 4).
> > > The neural dataset from the monkey OFC is representative of the kind of complex biological data available to neuroscientists at the moment. It is important to note that we do not have ground truth (labels) available for this data - we do not know what the OFC encodes (it is one of the most complex regions and neuroscientists currently do not have a good understanding of its function). As such, we do not know a priori whether OFC neural signals encode the information in such a way so as to allow us to perform classification between the different periods or between match and mismatch conditions. We find that TRAKR is able to distinguish three different periods, at relatively higher accuracy than other approaches. The finding that classification is possible is a novel result, and the result that TRAKR does it with high accuracy we believe makes this an important tool for the (neuro)scientific community to assess the kind of information that is encoded in complex biological systems.
> > > As far as Fig 5, we agree with the reviewer that a comparison would be useful, and we are planning to include one in the final submission. We have moved this figure to the Appendix for now and revised the accompanying explanation to indicate how projections into the space of the reservoir can allow for comparing the similarity of various sequences visually, thus generating useful hypotheses for further analyses or experiments.

---

> > ### Comment · Reviewer_xpen · 2021-11-29
> > **Authors' response clarified a lot of details and improved the paper**
> >
> > The authors addressed my remarks and significantly improved the quality of the paper.
> > I appreciate the update in Figure 3 and Table 1, and a lot of detailed have been clarified.
> > I changed my score from 5 to 6.
> >
> > The addition of ESN and MLP methods to the benchmark is a nice addition. I think the paper would also benefit from a comparison with a random projection technique (e.g. random Fourier features + SVM) which should be a fast and decent baseline. If TRAKR outperforms this baseline, it would demonstrates the benefit of TRAKR learning features specific to the training dataset.
> >
> > I think Table 1 could be further improved to convey more intuition than a list of numbers. For instance, Table 1 could be moved to the appendix for completeness, and a new Figure could be created instead:
> > - scatter plot, one dot per method (or two if the method has a separate projection and classifier)
> > - x-axis = inference time (in log-scale)
> > - y-axis = AUC or accuracy
> > This would demonstrate more intuitively the fact that TRAKR is not the best accuracy nor the fastest, but has the best combined performances. I think the authors should explore this sort of plot to see if it intuitively conveys their main message. The figure could also have the training time, either in the same plot or in a separate plot.

---

> ### Author Response · Authors · 2021-11-22
> **Manuscript Revision Uploaded**
>
> We thank the reviews for their time and insightful comments. We have uploaded a revised manuscript based on their comments. Concretely, the following updates have been made:
>
> - The abstract has been revised based on reviewer suggestions, and an updated manuscript provided.
> - We compared the performance of TRAKR with 2 additional supervised learning methods: a deep learning based approach (multi layer perceptron (MLP) as in Fawaz et al 2019) and an echo-state network approach (time warping invariant echo state network (twiESN) as in Fawaz et al 2019). We show that TRAKR performs on par with MLP and outperforms the previous ESN approach in both the sequential MNIST dataset (Figures 3A and 3B) as well as the macaque OFC neural recordings (Figures 4C and 4D). We also revised the text related to these in sections 3.2, 3.3 and 4.
> - Updated Figures 3A, 3B, 4C and 4D as mentioned and updated text in the Results section to reflect those changes. Moved former Figure 5 to the appendix and revised the accompanying text at the end of Results.
> - Updated all Figure captions
> - We revised Table 1, which now shows both training and inference times for all methods, including the train and inference time for the SVM classifier used for TRAKR and other approaches.
> - We re-ran the analyses with a different DTW implementation (based on Rakthanmanon et al 2012) which is described as faster and more accurate than the previously used implementation (FastDTW). We adjusted the captions of Figures 3 and 4 to reflect this change.
> - In response to reviewers’ comments, we now used 10-fold stratified cross validation throughout, instead of leave-one-out cross validation as before, as further explained in section 2.4.
> - Moved some details to the appendix sections; specifically:
> -- Details on the length of macaque trials
> -- Data preprocessing of macaque neural signals
> -- Details on the noise added to seq- MNIST digits
> -- Former figure 5 and its caption as text

---

### Official Review · Reviewer_m9nG · 2021-11-01

**Correctness:** 3
**Technical Novelty And Significance:** 2
**Empirical Novelty And Significance:** 2
**Recommendation:** 3
**Confidence:** 4

**Main Review:**

Strength
1. The paper is clearly written and easy to follow.
2. The proposed approach is simple, easy to implement, and already provides a gain over conventional methods.

Weaknesses
1. The technical contribution is unclear, since reservoir computing is not new. Would be interesting to see if we learn an embedding space (using simpler methods) with a given set of reference time segments and use that embedding to reconstruct other segments to generate error signals, whether we could achieve the same gain as reservoir computing. This experiment can help highlight the need for reservoir computing.
2. It is unclear what would be a good choice of “reference” time segments/sequences. For ECoG, rest period would make sense, but is there a way to decide in the general setting? Is the contribution of this work actually the use of a reference to generate error signals for classification, which tends to increase accuracy?
3. It is unclear what the simulated timeseries experiment is demonstrating. If we overfit the training time segments, by construction, we would get huge error signals on test segments, which would enable us to distinguish the two.
4. For the sequential MNIST experiment, it is not clear which digit would be a good reference and how results change with this choice. Nevertheless, getting an AUC of 99% with such a simple approach is quite impressive. If we use 10-fold cross validation, does the high AUC stays? Sometimes leave-one-out provides error estimates similar to training errors.
5. Though TRAKR is fitted in one shot, does performance improve with a few more rounds? In general, when deploying a model real time, we care more about the amount of time it takes to generate a prediction from the model, and not so much its training time (unless the amount of training time is impractical). Hence, comparisons with other DL models are needed to show TRAKR’s performance is at least on par.
6. Overall, seems like TRAKR works well for relatively simple classification tasks but breaks down on harder tasks like match vs. mismatch.


**Summary Of The Paper:**

This paper proposes using reservoir computing for classifying time segments. The key advantage is less training time. The approach, TRAKR, is evaluated on simulated timeseries, sequential MNIST, and ECoG with increased accuracy over conventional methods shown.

**Summary Of The Review:**

My score is mainly based on TRAKR not evaluated on harder classification tasks, lack of comparison with state-of-the-art DL models, and unclear technical novelty.

Post-revision summary
The authors addressed some of my comments. However, the motivation as written remains to be faster runtime for real world applications, but accuracy is lower than MLP and runtime is not that much shorter. A little surprising, but would expect the state-of-the-art to be some form of transformers as opposed to MLP. Given MLP provides higher performance, I will keep my score.

---

> ### Author Response · Authors · 2021-11-22
> **Response to reviewer's comments**
>
> We thank the reviewer for taking the time to review and the suggestions. We will address them individually below.
> 1) We appreciate this suggestion. As Fawaz et. al. (2019) - a comprehensive review on deep learning methods  for time series classification - points out, more powerful/ computationally intensive approaches such as DNNs (deep neural networks) are able to achieve good results that “are not significantly different” from much more computationally intensive approaches such as ‘COTE’ (an ensemble method that “requires training 37 different classifiers”).
> This suggests there is a need and use case for neural network based approaches. Departing from this, we hypothesized reservoir computing might perform similarly to DNNs while requiring much less training and inference time with the right approach.
> Our contribution is to offer a reservoir computing approach that is competitive with state-of-the-art methods in time series classification, such as explained further in (5) and (6) below, while being much less compute-heavy compared to approaches such as the ones mentioned above. To achieve this, we have fitted the reservoir to single time series trajectories and derived an error signal which we use as a measure of similarity. Those two things we found are crucial in yielding the high performance we achieve on the sequential MNIST and the neural dataset we compare all approaches on.
>
> 2) As the reviewer correctly points out, using a reference to generate the error signal is indeed crucial to our approach (as further explained in 1). In response to the other reviewers’ comments, we have now also compared our approach to a previous reservoir-based approach - time warping invariant ESN (Tanisaro & Heidemann, 2016) - which is also employed as one of the comparisons in the review paper above (Fawaz et. al. 2019). We find that our approach outperforms time-warping invariant ESN, which achieves ~73% Accuracy on the sequential MNIST data.
> Currently, we do not handpick a particular reference, but instead average our results (Figs 3 & 4) across all possible references. To be fair in comparison, we also do this with other distance metrics and measures such as DTW (which is non-symmetric).
> In the future, the selection of a particular reference signal could be treated as a hyperparameter, to be optimized on a smaller validation set. It is important to note, though, that one can already achieve good results by picking a reference ad hoc: the variance across different references is relatively small (TRAKR error bar, Figs 3 & 4).
>
> 3) The synthetic time-series experiment serves as a sanity check, before applying our method to real-world data. We intentionally want to overfit a particular signal to see the effect on the error signal when compared to other signals of varying frequency and amplitude. We have systematically explored this, but only included this figure for the sake of brevity.
>
> 4) We appreciate the reviewer’s remark. As explained above (in 2), we currently average our results across all possible references for the sake of a fair comparison. The variation across different references is pretty small, though, and we can achieve results >95% with any particular reference. In the future, one can also treat the particular reference to be selected as a hyperparameter.
> We agree that it is a good result for such a simple approach. This is in fact the main motivation for our contribution: good results coupled with ease-of-use/low compute. We believe these factors are of high importance for end-users, especially in the medical field, who face significant time constraints.
> We appreciate the suggestion of using a different number of folds. We have now run the analysis with 10-fold cross validation as well, and find that the results remain the same. We have now re-run all analyses with 10-fold cross validation and revised all reported results (Figs 3 & 4).

---

> > ### Author Response · Authors · 2021-11-22
> > **Response to reviewer's comments**
> >
> > 5) We appreciate the reviewer’s suggestion. It is important to note that the readout weights are ‘forced’ to fit the desired pattern in one shot, thus obviating the need for multiple rounds of optimisation on the same pattern. We see this as one of the main benefits/strengths of our approach, compared to other approaches such as DNNs (which require many rounds of optimisation) or COTE (which require training 37 different classifiers, as further explained in 1 above). Overall, training time is significantly reduced with TRAKR. While we agree that inference time is crucial, we also believe that training time is of concern in many real-world applications, especially in the medical realm: recording electrodes degrade/move with time and biological signals shift, requiring constant re-training of classifiers. We make this more clear in the revision.
> > The question whether we could achieve further performance gains by training on more examples of a given reference pattern is an interesting one, and we are exploring this currently. So far we find that using more examples from relatively more simple, repetitive patterns does not yield significant improvements, while it can have a relatively higher impact when patterns are complex (e.g. traces generated from chaotic recurrent networks).
> > Upon the reviewer’s suggestion, we have compared our approach to other ESN and DNN models. Comparing our approach to time warping invariant ESN (Tanisaro & Heidemann, 2016), as also further explained in 2, we find it yields lower performance (73% Accuracy) compared to our approach, at relatively higher inference time (~250ms vs. 40ms for TRAKR alone without the SVM). Comparing to MLP (multi-layered perceptron), using the same configuration as in Fawaz et al. (2019), we find it achieves similar performance at higher inference time (100ms vs. 40ms for TRAKR alone without SVM). Also, we found training time for MLP to be much higher compared to TRAKR(110s for MLP vs. 15s for TRAKR with SVM). We have now updated Figs 3 & 4 and the accompanying description in the Results section to reflect these new insights. Although we have used SVM for the classification of error signals here, other relatively more simple classifiers such as kNN (k-Nearest Neighbors) may also be used instead, for further gains in speed, as we now point out in the Discussion section. Overall, we believe these results will have a true impact for practical use of TRAKR for time-series classification tasks in a real-world setting.
> >
> > 6) We would like to stress that the classification tasks in either of the two real-world datasets we employ is not straightforward. Dynamic time warping (DTW) is a state-of the art technique for the classification of time series signals (“DTW when used with a NN classifier has been shown to be a very strong baseline” (Fawaz et al. 2019; Bagnall et al 2017); “At a minimum, it is clear that 1NN-DTW will get you within 98% of the best accuracy possible, in the first five minutes” (Eamonn Keogh, https://www.cs.unm.edu/~mueen/DTW.pdf)). We found this technique, in its optimised implementation (based on Rakthanmanon et al 2012), to yield lower results than TRAKR on both datasets.
> > The neural dataset from the monkey OFC is representative of the kind of complex biological data available to neuroscientists at the moment. It is important to note that we do not have ground truth (labels) available for this data - we do not know what the OFC encodes (it is one of the most complex regions and neuroscientists currently do not have a good understanding of its function). As such, we do not know a priori whether OFC neural signals encode the information in such a way so as to allow us to perform classification between the different periods or between match and mismatch conditions.
> > We find that TRAKR is able to distinguish three different periods, at relatively higher accuracy than other approaches. The finding that classification is possible is already a novel result, and the result that TRAKR does it with high accuracy makes this an important tool for the (neuro)scientific community to assess the kind of information that is encoded in complex systems.
> > The fact that TRAKR as well as all other approaches (including MLPs) are not able to differentiate between match and mismatch trials does not suggest that TRAKR is at fault, but rather that the neural signal in the OFC does not contain enough information to distinguish the two.

---

> ### Author Response · Authors · 2021-11-22
> **Manuscript Revision Uploaded**
>
> We thank the reviews for their time and insightful comments. We have uploaded a revised manuscript based on their comments. Concretely, the following updates have been made:
>
> - The abstract has been revised based on reviewer suggestions, and an updated manuscript provided.
> - We compared the performance of TRAKR with 2 additional supervised learning methods: a deep learning based approach (multi layer perceptron (MLP) as in Fawaz et al 2019) and an echo-state network approach (time warping invariant echo state network (twiESN) as in Fawaz et al 2019). We show that TRAKR performs on par with MLP and outperforms the previous ESN approach in both the sequential MNIST dataset (Figures 3A and 3B) as well as the macaque OFC neural recordings (Figures 4C and 4D). We also revised the text related to these in sections 3.2, 3.3 and 4.
> - Updated Figures 3A, 3B, 4C and 4D as mentioned and updated text in the Results section to reflect those changes. Moved former Figure 5 to the appendix and revised the accompanying text at the end of Results.
> - Updated all Figure captions
> - We revised Table 1, which now shows both training and inference times for all methods, including the train and inference time for the SVM classifier used for TRAKR and other approaches.
> - We re-ran the analyses with a different DTW implementation (based on Rakthanmanon et al 2012) which is described as faster and more accurate than the previously used implementation (FastDTW). We adjusted the captions of Figures 3 and 4 to reflect this change.
> - In response to reviewers’ comments, we now used 10-fold stratified cross validation throughout, instead of leave-one-out cross validation as before, as further explained in section 2.4.
> - Moved some details to the appendix sections; specifically:
> -- Details on the length of macaque trials
> -- Data preprocessing of macaque neural signals
> -- Details on the noise added to seq- MNIST digits
> -- Former figure 5 and its caption as text

---

### Decision · Program_Chairs · 2022-01-20

**Decision:**

Reject

**Comment:**

Authors propose an autoencoding echo state machine for a one-shot one-class time series classification task. Their approach feeds a (one-dimensional) error signal over time relative to a reference training datum to SVMs. Training is very fast by design. OFC signal analysis has practical value in neuroscience. But only one benchmark (seq-MNIST) was used to evaluate their method. While the performance seem impressive, no explanation of why the internal representation learned by the proposed system is superior and robust to noise was provided. No sequential autoencoders or latent neural trajectory inference methods were compared. Although the manuscript has greatly improved through the review-rebuttal process, there are missing key details (e.g. length of E(t) used for classification--important for real-time application, initial state for the reservoir, choice of W_in --important since it seems to be a chaotic network that's driven by strong input). While there is novelty in the approach, there is a general lack of enthusiasm among the reviewers for the manuscript as is. The reviewers and AC strongly encourage the authors to further developed these ideas and add thorough analyses for another conference.

(BTW, perhaps it's worth citing https://doi.org/10.1109/IJCNN.2016.7727309, since autoencoder combined with reservoir computing has been used for anomaly detection.)